# The impact of international megaproject social responsibility on satisfaction with the environmental compensation mechanism: The role of stakeholder participation

Zixuan Zeng[1]*, Yongtao Shen[1], Jijun Yang[2], Thammanoon Hengsadeekul[3]

**1** Guangxi Vocational and Technical Institute of Industry, Nanning, Guangxi, China, **2** School of Finance, Guangdong University of Finance and Economics, Guangzhou, Guangdong, China, **3** Faculty of Logistics and Digital Supply Chain, Naresuan University, Phitsanulok, Thailand

* zengzixuan@gxgy.edu.cn

## Abstract

### Background

Construction projects, particularly megaprojects, inherently entail substantial environmental and social challenges. Early research on megaproject social responsibility (MSR) has primarily focused on domestic contexts, such as China, and has explored resident satisfaction mainly through economic or policy lenses. However, these studies have paid limited attention to broader stakeholders, leaving a significant gap in understanding how MSR influences satisfaction with the environmental compensation mechanism (SECM), particularly in international settings. This study aims to fill this gap by examining how MSR at different stages of the project life cycle affects SECM and by analyzing the mediating role of stakeholder environmental activities participation (SEAP) in this relationship. The findings provide theoretical and empirical insights to enhance the design and implementation of environmental compensation in international megaprojects, thereby contributing to improved stakeholder satisfaction and sustainable development outcomes.

### Methods

Grounded in life-cycle and stakeholder theories, this study investigated large-scale projects involving Chinese contractors in Thailand. We employed a two-stage questionnaire survey, beginning with a pilot study (n=204) to validate the scales, followed by a main study (n=398) for final analysis. Data were analyzed using factor analysis, Pearson correlation, and multiple linear regression.

**Data availability statement:** All relevant data are within the manuscript and its Supporting Information files.

**Funding:** This study is supported by the National Natural Science Foundation of China (12261006) and Middle-aged and Young Teachers' Basic Ability Promotion Project of Guangxi (2022KY0647, 2022KY0640). The funders had no role in study design, data collection and analysis, decision to publish, or preparation of the manuscript.

**Competing interests:** The authors have declared that no competing interests exist.

## Results

The results indicate that MSR has a significant positive effect on host-country residents' satisfaction with the environmental compensation mechanism (SECM), with direct effects notably originating from the design and operational stages. Furthermore, stakeholder environmental activities participation (SEAP) partially mediates the relationship between MSR (across all its stages) and SECM.

## Conclusions

This study enriches the literature by providing transnational empirical evidence on the role of MSR in improving SECM. It highlights the critical importance of the design and operation stages and confirms the mediating role of stakeholder participation. Practically, the findings offer valuable insights for international project managers and policymakers, emphasizing the strategic importance of engaging stakeholders in environmental activities to achieve sustainable development goals and enhance community satisfaction.

## Introduction

Megaprojects are complex, large-scale engineering ventures that provide essential public services, contributing to social production, economic development, and livelihoods. They involve multiple stakeholders and exert a profound impact on societies and states; consequently, they cannot be regarded merely as scaled-up versions of smaller projects [1]. Globally, megaprojects are advancing at an unprecedented scale and pace, characterized by massive capital investments and extended construction timelines [2]. They often cross national borders and exert profound impacts on the natural environment, social structures, and economic landscapes of host regions and surrounding areas [3]. For instance, global investment in large-scale infrastructure amounts to trillions of U.S. dollars annually, triggering significant transformations in local ecosystems, land use patterns, and community livelihoods. In this context, external forces—such as the United Nations Sustainable Development Goals (SDGs), increasingly stringent environmental regulations in host countries, and the environmental and social safeguard policy frameworks of international financial institutions like the World Bank—collectively drive international megaprojects to prioritize and actively fulfill their social responsibilities.

In this context, ensuring environmental sustainability and obtaining a social license to operate from local communities present critical challenges for project proponents. In response, International Megaproject Social Responsibility (MSR) has emerged as a critical pathway for ensuring project legitimacy, mitigating operational risks, and achieving long-term sustainable development [4]. MSR requires project proponents to actively identify, manage, and respond to the environmental and social impacts of a project throughout its entire life cycle, from initiation and design to construction, operation, and ultimately decommissioning [5,6]. Particularly concerning environmental

damage caused by projects, environmental compensation mechanisms represent a core measure for mitigating negative ecological consequences and restoring affected resources [7]. However, although widely applied, the effectiveness of these mechanisms and the satisfaction of affected stakeholders often fall short of expectations [8,9]. This deficiency not only hinders the achievement of compensation goals but can also incite community resistance and result in significant project delays [10]. While the existing literature has separately explored topics such as MSR, environmental impact assessment, environmental compensation, and stakeholder participation, a significant research gap remains: a lack of systematic, in-depth investigation into how MSR specifically influences satisfaction with environmental compensation mechanisms (SECM). Crucially, the role that stakeholder participation in environmental activities (SEAP) plays in this relationship is particularly underexplored. Moreover, much of the research on megaprojects has been conducted in domestic contexts, with relatively few studies undertaken from a transnational or cross-cultural perspective [11]. This study seeks to address these research gaps from both theoretical and empirical perspectives. It aims to investigate the pathways through which international MSR influences satisfaction with the environmental compensation mechanism (SECM). Specifically, its primary objective is to examine the direct effect of MSR—including its manifestations across different stages of the project life cycle—on SECM. Second, it aims to explore the mediating role played by stakeholder environmental activities participation (SEAP) in the relationship between MSR and SECM. By addressing these objectives, this study will provide important insights for international contractors seeking to enhance their market understanding, competitiveness, and sustainable outcomes in complex cross-border projects.

## Literature review and research hypothesis

Megaproject Social Responsibility (MSR) refers to the policies and practices adopted by stakeholders involved throughout the entire life cycle of large-scale infrastructure projects. These are designed to address and balance the projects' economic, environmental, political, and ethical impacts on society [6,12]. Since its conceptual origin, research on MSR has evolved from initially discussing social and financial impacts [13], to addressing systemic challenges such as cost overruns and risk mismanagement [14], and finally to encompassing comprehensive assessments of community, environmental, and public welfare outcomes [15,16].

In China, although MSR scholarship is a more recent development, it has drawn on international research to construct localized organizational models and governance frameworks [16,17], as well as tailored indicator systems for measuring MSR performance [18]. International experience and comparative studies indicate that MSR implementation in China is shaped by a confluence of regulatory, ethical, economic, and political factors [7]. Recent research has further specified these as four core dimensions: economy, politics, ethics, and policy. This framework allows for a holistic perspective on the responsibilities assumed by various stakeholders throughout the megaproject life cycle [6].

Theoretically, MSR draws on stakeholder theory and institutional theory to explain how project actors negotiate social expectations, comply with regulations, and pursue legitimacy [4,5]. The dynamic relationships among stakeholders—including government bodies, market actors, contractors, local communities, and NGOs—are recognized as crucial in shaping both the evolution and outcomes of MSR practices. Empirical studies suggest that robust MSR implementation is associated with multiple positive outcomes. For example, Ma and Sun [19] found that effective MSR enhances both the financial and social performance of organizations participating in megaprojects. In a biodiversity case study during the construction of the Hong Kong-Zhuhai-Macao Bridge, Liu et al. [7] showed that carefully planned ecological compensation led to an increase, rather than a decline, in the local Chinese white dolphin population. MSR has also been reported to help reduce transaction costs and operational risks, while fostering sustainability within the construction industry [5,20]. Additionally, MSR can drive cost savings, foster employee motivation, and enhance organizational reputation [21].

Conversely, inadequate or superficial MSR can have adverse effects. Problems such as administrative overreach, loss of public trust, opportunistic behaviors, or even corruption and bribery may undermine project legitimacy and community acceptance [6]. Lin and Zeng [22] demonstrated that environmental irresponsibility damages corporate reputation, with

social responsibility activities only partially mitigating these effects. In the absence of meaningful social responsibility, negative outcomes may include unresolved community resettlement, ineffective pollution control, ecological degradation, occupational safety incidents, and weakened anti-corruption efforts [5,23].

MSR is now widely considered a key component of risk management, stakeholder satisfaction, and sustainable development in megaprojects. It requires both context-sensitive design and rigorous implementation across all project stages [24].

The implementation of international megaprojects often generates significant environmental impacts. Consequently, environmental compensation mechanisms constitute a crucial link for projects to obtain a social license to operate and ensure long-term sustainability [25]. Environmental compensation aims to mitigate ecological damage caused by projects through measures such as restoration, reconstruction, or the provision of alternative resources [26]. However, the ultimate effectiveness of these mechanisms is reflected not only in the degree of ecological restoration but, more critically, in the satisfaction of affected stakeholders—particularly local community residents—with both the compensation process and its outcomes [27,28]. This stakeholder satisfaction is vital because it directly influences attitudes toward the project, affects the potential for conflict, and impacts the project's overall progress [29]. For instance, local communities tend to develop a favorable perception of large-scale hydropower projects if they receive adequate compensation for property expropriation (particularly in cases involving displacement) or if they anticipate opportunities for livelihood enhancement.

Although the development of international megaprojects can potentially improve socio-economic standards in host communities [10]—as observed with biomass power plants and sugar mills in southern Thailand that improved local electricity access and waste recycling—the inherent characteristics of megaprojects, such as substantial resource consumption, long life cycles, and diverse stakeholders, often induce significant changes in the social environment. Adriaanse [30] emphasized that changes in the social environment of residential areas are a key determinant of satisfaction. Such changes can result in stakeholder dissatisfaction and even contribute to project failure [31]. MSR provides a framework for responsible behavior designed to address these complex impacts and mitigate potential dissatisfaction. A high level of MSR prompts project proponents to be more proactive in environmental management, to adopt prevention and mitigation measures, and to design and implement environmental compensation schemes more responsibly. These MSR practices help reduce the environmental and social problems that cause dissatisfaction, enhance the effectiveness and perceived fairness of environmental compensation measures, and thereby positively influence the perceived satisfaction of affected groups.

A deeper examination into the concept of MSR across the project life cycle stages allows for a more precise understanding of its specific pathways for influencing environmental compensation satisfaction. In the initiation and design stages, MSR is reflected in the prioritization of environmental factors in early project decisions, the conduction of comprehensive environmental impact assessments (EIA), the identification and prediction of potential environmental risks and affected groups, and the subsequent design of scientific and forward-looking environmental compensation plans. Responsible practices at this stage can effectively prevent or mitigate future environmental problems and lay the foundation for rational and feasible compensation plans, thereby indirectly enhancing eventual compensation satisfaction [26]. During the **construction stage**, MSR requires proponents to strictly adhere to environmental management plans, implement effective pollution control measures, minimize disturbance to the surrounding environment and communities, and promptly initiate restoration or compensation for any unavoidable environmental damage [32,33]. Responsible construction behavior directly influences affected groups' immediate perceptions of the project's environmental performance and compensation implementation during this critical period, consequently impacting their satisfaction.

In the **operation stage**, the environmental impacts of large projects may become long-term or cumulative. MSR at this stage requires proponents to continuously monitor environmental conditions, maintain environmental facilities, strictly control operational emissions, and fulfill all long-term environmental compensation or ecological restoration commitments [34]. Sustained responsible behavior during the operation phase is crucial for maintaining and enhancing affected the long-term

satisfaction of groups with the environmental compensation mechanisms [35]. The form and focus of MSR vary across the project life cycle stages, and their influence on environmental compensation satisfaction also differs in mechanism and intensity. Together, these stage-specific effects form a complex link from project responsibility to compensation satisfaction [16]. Based on the above analysis of MSR's influence at each stage, we hypothesize that MSR across the different life cycle stages positively influences satisfaction with the environmental compensation mechanism. Therefore, we propose the following hypothesis:

H1: MSR positively influences satisfaction with the environmental compensation mechanism (SECM).

H1a: The initiating stage of MSR (MSR-IS) positively influences SECM.

H1b: The design stage of MSR (MSR-DS) positively influences SECM.

H1c: The construction stage of MSR (MSR-CS) positively influences SECM.

H1d: The operation stage of MSR (MSR-OS) positively influences SECM.

High-quality stakeholder participation and effective information-collection tools are key elements for the success of megaprojects, such as in the development of transport plans [36]. Li and Thomas Ng [37] illustrated that the level of stakeholder participation in megaprojects was limited in China, especially during the earlier stages, primarily due to cultural and value differences, unbalanced participation mechanisms, and a lack of confidence. Furthermore, stakeholder participation can serve as a means to resolve conflicts and understand the positions of the parties involved [38]. This has been demonstrated in conflict analyses of stakeholder participation in Hong Kong urban planning projects [39] and in studies of conflicts arising from environmental issues during the construction of the Hong Kong-Zhuhai-Macao Bridge [40]. Meanwhile, MSR can help participating organizations build relationships with stakeholders and gain economic, political, and social legitimacy [41]. Stakeholder participation can improve the quality of environmental decision-making by incorporating more comprehensive information inputs, although empirical evidence supporting this claim remains somewhat limited [42]. Thus, we propose Hypothesis 2:

H2: MSR is positively related to stakeholder environmental activities participation (SEAP).

As the project life cycle progresses, the attributes of key stakeholders and the central issues of social responsibility change dynamically [43]. In the initiating stage, key stakeholders are governments and local communities. Gaining public acceptance and ensuring proper disclosure of information at this stage are of great significance for the subsequent stages of the megaproject [6]. For example, public hearings were found to have moderate effectiveness when conducted prior to the start of megaprojects in Thailand [44]. In the design stage, following acceptance by top management, more detailed plans are established to accomplish initial goals. During this stage, the government should adhere to industrial standards and promote green innovation technologies, while designers should actively incorporate green design principles [6,45]. Additionally, the public tends to focus on the long-term effects of megaprojects while their priorities remain dependent on the local socio-economic situation [46]. For example, in the case of a bridge in the United States, the public may emphasize aesthetics over advanced technology due to the structure's historical significance [47]. In contrast, for a new bridge project, the public may be less concerned with aesthetics and more concerned with its impact on local businesses, job markets, and other socio-economic factors [48]. During the construction phase, sufficient materials and resources are procured to achieve the desired project outcomes [49] and all stakeholders are involved in diverse ways. At this stage, the process of public participation is often one-way, as the public typically cannot directly engage with the project and primarily receives construction-related information from it, such as notices about road closures or safety hazards [46]. In the operation stage, where the majority of a building's carbon emissions are typically generated, NGOs and district councils

can assume a greater role in continuously monitoring the social and environmental impacts of megaprojects, especially after primary stakeholders have exited the project [50]. Thus, responses to public events and the maintenance of environmental balance present unique challenges and features during this project stage [6]. Hence, we propose the following sub-hypotheses for H2:

H2a: The initiating stage of MSR (MSR-IS) is positively related to SEAP.

H2b: The design stage of MSR (MSR-DS) is positively related to SEAP.

H2c: The construction stage of MSR (MSR-CS) is positively related to SEAP.

H2d: The operation stage of MSR (MSR-OS) is positively related to SEAP.

Research on stakeholder participation suggests that training can enhance satisfaction. For instance, from the perspectives of both staff training and community residents, participants who receive effective training on building functions report higher satisfaction with their environment compared to those who do not receive such training [51].The adoption of energy-efficient design strategies in the construction industry attracts attention due to concerns about corporate policies, government regulations, cost-effectiveness, utility incentives, energy use reduction goals, and occupant productivity and satisfaction [52,53]. Furthermore, the success of these strategies largely depends on the interaction between residents and the construction process [54]. Economic benefits derived from megaprojects have a positive effect on community satisfaction, whereas negative environmental impacts typically have a negative effect. However, the participation of social media can help create a more positive perception of megaprojects [10]. From the perspective of economic development, Cai and Yu [55] assessed the impact of policy effectiveness on rural environmental compensation satisfaction through questionnaire surveys conducted in Chengdu, Suzhou and Shanghai in 2012 and 2015. Their results showed that a quicker policy response time correlated positively with higher satisfaction among the rural population, though satisfaction levels varied across the different cities. Farmers in Chengdu reported higher satisfaction levels than those in Suzhou and Shanghai, suggesting that local economic development may influence farmers' evaluations of compensation policies. From a social equity perspective, environmental compensation can play an important role in protecting social equity, as it directly links compensation to community interests, particularly through the promotion of local livelihoods [56]. Although environmental management constitutes a part of social responsibility, the implementation of environmental compensation is still fraught with conflicts [57] and risks [58]. Identifying and implementing effective measures to manage these conflicts and risks remains a critical issue for exploration. Based on this reasoning, we propose Hypotheses 3 and 4, along with the corresponding sub-hypotheses (4a-4d):

H3: Stakeholder environmental activities participation (SEAP) is positively associated with satisfaction with the environmental compensation mechanism (SECM).

H4: Stakeholder environmental activities participation (SEAP) mediates the positive relationship between MSR and satisfaction with the environmental compensation mechanism (SECM).

H4a: SEAP mediates the positive relationship between the initiating stage of MSR (MSR-IS) and SECM.

H4b: SEAP mediates the positive relationship between the design stage of MSR (MSR-DS) and SECM.

H4c: SEAP mediates the positive relationship between the construction stage of MSR (MSR-CS) and SECM.

H4d: SEAP mediates the positive relationship between the operation stage of MSR (MSR-OS) and SECM.

In summary, based on the literature review, this study establishes four main research hypotheses and twelve sub-hypotheses. The conceptual model summarizing these hypotheses is presented in Fig 1.

## Methodology

This study employed a mixed-methods research approach, which progressed through several interconnected stages. First, a comprehensive review of existing literature was conducted to establish the theoretical foundation and identify gaps in current knowledge. Based on these insights, preliminary qualitative research was conducted in the form of semi-structured interviews to gather contextual information and refine the research focus. Subsequently, a quantitative survey was designed and administered to collect empirical data from relevant stakeholders. The collected data were then analyzed to examine the relationships among key variables and to test the proposed hypotheses. Finally, the findings were interpreted and discussed in light of the theoretical framework and prior studies, leading to the conclusions and implications of the research. The overall aim of this process was to clarify how megaproject social responsibility influences satisfaction with environmental compensation mechanisms and to elucidate the mediating role of stakeholder participation in this relationship.

### Research context and design

This research was situated in the context of international megaprojects involving Chinese contractors in Thailand. This specific context was selected for its unique theoretical and practical value, which includes the increasing presence of Chinese contractors in Thailand's infrastructure sector, the distinct cross-cultural stakeholder dynamics inherent in such projects, and the need to better understand MSR effectiveness and environmental outcomes within a specific developing country context. Data for the main quantitative study were collected through a structured questionnaire administered to key stakeholders affected by the projects, including those associated with the Eastern Sugar & Cane Public Company Limited in Wang Sombun, Sa Kaeo, and the PTG 24MW Biomass-fired Power Plant Project in Nong Chik, Pattani.

The overall research process followed a systematic design procedure as illustrated in Fig 2. This process commenced with a comprehensive literature review to build the theoretical framework and guide initial conceptualization. As

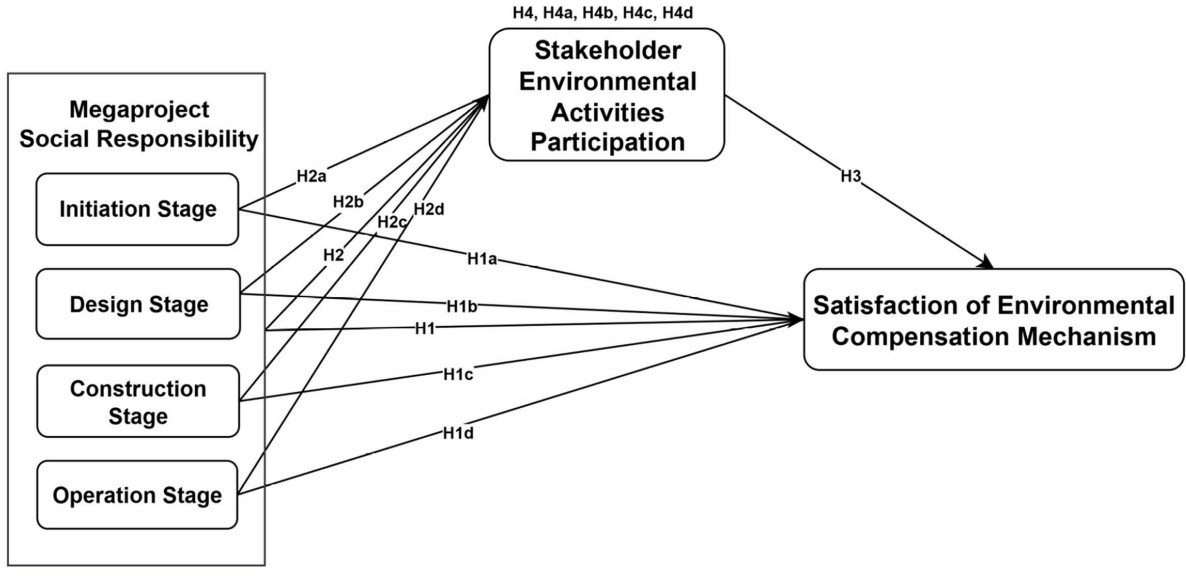

**Fig 1. Conceptual Model.**

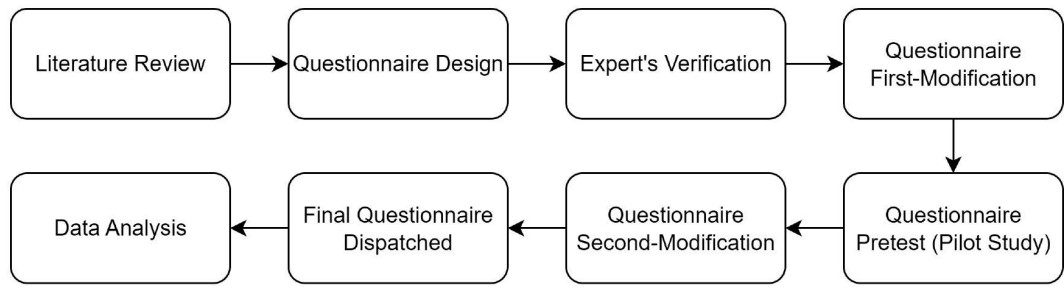

**Fig 2. Research Process.**

a preliminary step to inform the survey design and enhance the understanding of the research context—particularly the stakeholder structure of sugar mills in Thailand—semi-structured interviews were conducted. Participants included the General Manager, the Director of the R&D Department, and a Senior Engineer from Guangxi Construction Engineering Group No. 1 Installation Co., Ltd. The interview protocol, originally developed in Chinese, was translated into English using a back-translation technique to ensure linguistic equivalence [59]. The insights gained from these interviews, including a summary of the stakeholder structure, informed the subsequent development of the questionnaire and the overall research design.

To ensure the validity of the survey instrument, a rigorous development procedure was followed, as outlined in Fig 2. The questionnaire was developed based on constructs and items adapted from the existing literature. Specifically, the items measuring MSR were adapted from the scales developed by Lin and Zeng [60], Kraisornsuthasinee and Swierczek [61], and Virakul et al. [62]. The items for SEAP were drawn from the work of Lamm et al. [63]. Finally, the items for SECM were adapted from Li and Ng [64]. First, we thoroughly reviewed the literature and designed the first draft of the measurement scales carefully. Second, we conducted interviews with experts in project management, mathematics and statistics, environmental management, international business, and satisfaction research to further refine the hypothetical model and the questionnaire draft. Third, an Item-Objective Congruence (IOC) assessment was performed by five out of seven Thai professors to verify the content validity of the questionnaire. Fourth, prior to administering the final survey, a pilot study was conducted with one construction firm to verify the phrasing, clarity, and comprehensibility of each question. Finally, a second round of modifications was implemented based on the pilot feedback before distributing the final questionnaire.

To reduce the potential for common method bias(CMB), several ex ante procedural remedies were employed, as recommended in the literature [66]. To minimize potential informant bias, respondents with similar roles were sampled, and the confidentiality of all responses was strictly maintained. The questionnaire items were ordered under general topic sections rather than being grouped The final questionnaire comprised three parts: (1) respondents' demographic information; (2) respondents' evaluation of their degree of participation in environmental activities across the megaproject life cycle; (3) respondents' satisfaction with the environmental compensation mechanism.

## Sample and data collection

The study employed a purposive sampling method to ensure that respondents were directly relevant to the research objectives. The target population for the main quantitative survey comprised diverse stakeholders who met the following inclusion criteria: (1) being aged 18 years or older; (2) being directly involved in the planning, design, construction, operation, or oversight of the selected international megaprojects in Thailand, or residing in and being significantly affected by the project areas; and (3) being willing to provide informed consent and complete the questionnaire in full. The stakeholder categories included Chinese contractors and subcontractors, central or local government officials, designers, local subcontractors, project legal personnel, suppliers, non-governmental organizations (NGOs), media representatives, and

local community or public members. These categories were selected because contractors, subcontractors, and suppliers directly influence the implementation of social responsibility measures. Government officials and designers play leading roles in policy formulation and technical decision-making. Non-governmental organizations and media outlets act as intermediaries, undertaking monitoring, reporting, and promoting improvements in environmental and social outcomes. Community and public members are the direct recipients of environmental compensation measures and are thus directly affected by the environmental performance of the projects. Including perspectives from all these groups enabled a more comprehensive understanding of how megaproject social responsibility relates to stakeholder participation and satisfaction with environmental compensation mechanisms. This sampling approach ensured that data were collected from multiple perspectives closely related to the core constructs of the study: megaproject social responsibility, stakeholder environmental activities participation, and satisfaction with environmental compensation mechanisms. A total of 485 questionnaires were distributed. After data cleaning and refinement, 398 valid questionnaires were retained for final analysis, yielding a response rate of 86.9%. The final sample consisted of 81 Chinese contractors/sub-contractors, 21 central or local government officials, 21 designers, 12 local sub-contractors, 7 project legal personnel, 9 suppliers, 8 NGO representatives, 7 media representatives, and 211 community/public members. In terms of educational attainment, the respondents included 17 holding a Ph.D., 45 with a master's degree, 233 with a bachelor's degree, 56 with a diploma, 43 with a high school education, and 4 with less than a high school education. The detailed demographic information of the respondents is summarized in Table 1. For data analysis, SPSS version 26.0 was used. The mediation analysis was performed using the PROCESS Macro for SPSS (version 4.0) developed by Hayes [65], which employs bootstrapping with bias-corrected confidence intervals.

To further reduce the potential for common method bias (CMB) [66], the aforementioned ex ante procedural remedies were applied during questionnaire design and data collection. These included ensuring respondent anonymity and questionnaire confidentiality, separating measurement items for different constructs, and varying item wording. As previously mentioned, to minimize informant bias, respondents with similar roles were sampled, and confidentiality was assured. Prior to analysis, the dataset underwent cleaning to check for missing values, identify and address outliers, and ensure overall data quality.

## Variable measurement

All perceptual variables were measured using multi-item scales anchored on a five-point Likert format. The average score of the items representing each construct was calculated and used in the subsequent analysis.

## Dependent variable

Satisfaction with the environmental compensation mechanism (SECM) was measured by adapting the method used by the Ministry of Natural Resources and Environment (MNRE) and the scale developed by Li and Ng [64]. This study developed an eight-item scale to measure SECM capturing dimensions such as public environmental consciousness, participation in environmental protection activities, production process and manufacturing technology, energy efficiency, financing systems for ecological conservation, the use of alternative energy, tax policies for environmental protection, and scientific and theoretical support. Items were rated on a scale from 1 ("not satisfied at all") to 5 ("very satisfied").

## independent variable

Megaproject Social Responsibility (MSR) was measured by adapting scales from Lin and Zeng [60] and Kraisornsuthasinee and Swierczek [61]. Respondents were asked to evaluate their perception of the fulfillment of MSR across four project stages: the initiating stage (MSR-IS), design stage (MSR-DS), construction stage (MSR-CS), and operation stage (MSR-OS). An overall MSR score was calculated as the average of the responses across all stages and items. The anchor points ranged from 1 ("never") to 5 ("always").

**Table 1. Demographic Information of Respondents.**

| Questions | Options | Frequency | Percentage (%) |
|---|---|---|---|
| Your gender is? | Male | 235 | 59.0 |
| | Female | 163 | 41.0 |
| What's your religion | Buddhism | 178 | 44.7 |
| | Islam | 7 | 1.8 |
| | Christian | 5 | 1.3 |
| | No Religion | 126 | 31.7 |
| | Other | 82 | 20.6 |
| How many years have you worked in the megaproject industry? | 1-5 years | 124 | 31.2 |
| | 6-10 years | 28 | 7.0 |
| | More than 10 years | 20 | 5.0 |
| | I'm not in that line of work | 226 | 56.8 |
| Please specify your highest education? | Ph.D. | 17 | 4.3 |
| | Master | 45 | 11.3 |
| | Bachelor | 233 | 58.5 |
| | Diploma/College | 56 | 14.1 |
| | High school | 43 | 10.8 |
| | Under the high school | 4 | 1.0 |
| Are you familiar with social responsibility? | Never heard at all | 18 | 4.5 |
| | A little understanding | 206 | 51.8 |
| | Have clear understanding | 105 | 26.4 |
| | Know more about this | 50 | 12.6 |
| | Very familiar | 19 | 4.8 |
| Mass media such as newspapers and television | unselected | 126 | 31.7 |
| | selected | 272 | 68.3 |
| Internet | unselected | 161 | 40.5 |
| | selected | 237 | 59.5 |
| Training and learning | unselected | 215 | 54.0 |
| | selected | 183 | 46.0 |
| Social occasions | unselected | 221 | 55.5 |
| | selected | 177 | 44.5 |
| Others | unselected | 299 | 75.1 |
| | selected | 99 | 24.9 |
| For megaprojects, which of the following groups do you belong to? | Chinese contractor or Sub-contractor | 81 | 20.4 |
| | The central government or local government | 21 | 5.3 |
| | Designer | 21 | 5.3 |
| | Local sub-contractor | 12 | 3.0 |
| | Project legal | 7 | 1.8 |
| | Supplier | 9 | 2.3 |
| | Project supervision | 21 | 5.3 |
| | NGO | 8 | 2.0 |
| | Media | 7 | 1.8 |
| | Community or public | 211 | 53.0 |

### Mediating variable

Stakeholder environmental activities participation (SEAP) refers to the degree to which stakeholders engage in environmental activities. SEAP was measured using the scale developed by Lamm et al. [63]. Respondents were asked to rate the degree of stakeholder participation in various environmental activities on a scale from 1 ("never") to 5 ("always"). Example items include the use of environmentally friendly building materials, proper waste disposal in designated locations, cleaning transport vehicles before use on public roads, regular participation in environmental protection education, and the provision of targeted garbage bins in various areas.

### Control variables

Based on the existing literature, several control variables were incorporated into the analysis. These control variables included: religion (Buddhism, Islam, Christianity, no religion, other); familiarity with social responsibility (never heard at all, a little understanding, clear understanding, know more about this, very familiar); highest education level attained (PhD, master's degree, bachelor's degree, diploma/college, high school, less than high school); and years of work experience in the megaproject industry (1–5 years, 6–10 years, more than 10 years, not applicable).

### Data collection method and ethical considerations

This study did not require formal Institutional Review Board (IRB) approval, as it did not involve sensitive personal information, vulnerable populations, or interventions that could cause physical or psychological harm. Prior to participation, an informed consent statement was provided at the beginning of the questionnaire. The statement included the purpose and method of the research, assurance of anonymity and confidentiality, the voluntary nature of participation, and the right to withdraw at any stage without penalty. Participants were required to indicate their consent before proceeding to the survey questions, and the questionnaire was programmed to terminate automatically if they declined to participate. The data were used solely for research purposes.

### Data analysis methods

The data collected were statistically analyzed using SPSS version 26.0 and the PROCESS Macro for SPSS (version 4.0). Descriptive statistics were used to analyze the demographic characteristics of the participants and the study variables, and Pearson's correlation analysis was conducted to examine the relationships among international megaproject social responsibility (MSR), stakeholder environmental activities participation (SEAP), and satisfaction with the environmental compensation mechanism (SECM). The direct effects of MSR and its four project life cycle stages (initiating, design, construction, and operation) on SECM were tested using hierarchical multiple linear regression, controlling for demographic variables such as religion, years of work experience, highest level of education, familiarity with social responsibility, and role in the project. The mediating role of SEAP in the relationships between MSR (and its stages) and SECM was analyzed using PROCESS Macro Model 4 with 5,000 bootstrap resamples. The significance of the indirect effect was determined by whether the 95% bias-corrected confidence interval (CI) excluded zero, and the proportion of the total effect explained by the mediation pathway was calculated to assess the strength of the mediating effect.

## Results

### Validity and reliability

**Pilot study.** A pilot study was first conducted to test the preliminary construct validity of the scales and to refine the items through an exploratory factor analysis (EFA), using principal component analysis. The pilot study results indicated acceptable validity and reliability for the initial scales. However, as seven components were extracted and some items

exhibited low correlations with their intended constructs, 18 items were omitted from the original scales based on these results and expert feedback. This process resulted in a refined set of scales for the main study.

**Multi-item constructs of the final study.** Subsequently, an EFA was performed on the remaining 68 items of the final questionnaire to further confirm their construct validity, again using principal component analysis. As shown in Table 2, the Kaiser-Meyer-Olkin (KMO) values for all constructs were above 0.955, and Bartlett's test of sphericity was significant (p < .001), indicating strong correlations among the variables and their suitability for factor analysis [67]. As detailed in Table 3, the total variance explained for each construct ranged from 74.39% to 82.81%, substantially exceeding recommended threshold of 40% [68]. Furthermore, all item factor loadings were above 0.617, and all communality values were greater than 0.603, indicating that each item had robust explanatory power.

To further evaluate the convergent and discriminant validity of the constructs, a confirmatory factor analysis (CFA) was performed using SPSS (version 26.0). The overall model fit indices ($\chi^2$/df = 2.569, p < .001; TLI = 0.894; CFI = 0.902; IFI = 0.903; NFI = 0.850; RMSEA = 0.068) demonstrated an acceptable fit to the data. As presented in Table 4, all standardized factor loadings were above 0.6, the composite reliability (CR) values all exceeded 0.9, and the average variance extracted (AVE) values for MSR, SEAP, and SECM were 0.710, 0.742, and 0.801, respectively. All of these values were well above their recommended thresholds (0.4, 0.7, and 0.5 for loadings, CR, and AVE, respectively), thus confirming good construct reliability and convergent validity. Additionally, discriminant validity was assessed using the Fornell-Larcker criterion, as shown in Table 5. For each construct, the square root of its AVE (shown on the diagonal in Table 5) was greater than its correlations with any other construct (off-diagonal values). This indicates that each construct shares more variance with its own measures than with measures of other constructs, thereby establishing discriminant validity.

The reliability test results, also presented in Table 3, indicated excellent internal consistency for all measures. The Cronbach's alpha values for each dimension ranged from 0.972 to 0.988, and the overall Cronbach's alpha was 0.992 [69], greatly exceeding the recommended threshold of 0.7. The corrected item-total correlation (CITC) values ranged from 0.692 to 0.873, further confirming the high homogeneity of the items and their strong relationship with the overall construct.

## Hypothesis test

**Descriptive statistics and correlation analysis.** Table 6 presents the descriptive statistics and intercorrelations for all study variables. The means, standard deviations, and Pearson correlation coefficients are displayed. As shown, MSR, its four stages (MSR-IS, MSR-DS, MSR-CS, MSR-OS), SEAP, and SECM were all significantly and positively correlated with each other. All correlation coefficients were below 0.9, indicating that multicollinearity was not a critical concern for the subsequent regression analyses. The correlation results provided preliminary support for Hypotheses 2 and 3. Specifically, a significant positive correlation was found between MSR and SEAP (r = 0.848, p < .01), providing initial support for H2. Thus, H2 was supported. Similarly, significant positive correlations were found between each stage of MSR and SEAP: MSR-IS (r = .766, p < .01, supporting H2a), MSR-DS (r = .746, p < .01, supporting H2b), MSR-CS (r = .833, p < .01, supporting H2c), and MSR-OS (r = .870, p < .01, supporting H2d). Furthermore, a significant positive correlation was found between SEAP and SECM (r = .728, p < .01), thus supporting H3. These strong correlations suggest that maintaining MSR throughout a project's life cycle may generate positive spillover effects, influencing both economic

**Table 2. The KMO value and significance.**

|        | KMO   | Chi-square | Sig.  |
|--------|-------|------------|-------|
| MSR    | 0.978 | 14,068.062 | 0.001 |
| SEAP   | 0.963 | 5117.108   | 0.001 |
| SECM   | 0.955 | 3916.904   | 0.001 |

**Table 3. Validity and reliability test results.**

| Variables | Construct/ Items | Total variance explained | Factor Loading | Communality | Corrected Item-Total Correlation | Cronbach's Alpha if Item Deleted | Cronbach's α |
|---|---|---|---|---|---|---|---|
| MSR | | 74.388 | | | | 0.985 | 0.992 |
| IS | IS1 | | 0.685 | 0.603 | 0.692 | | |
| | IS2 | | 0.730 | 0.704 | 0.765 | | |
| | IS3 | | 0.736 | 0.700 | 0.752 | | |
| | IS4 | | 0.728 | 0.693 | 0.748 | | |
| | IS5 | | 0.759 | 0.766 | 0.791 | | |
| | IS6 | | 0.730 | 0.714 | 0.765 | | |
| | IS7 | | 0.728 | 0.724 | 0.778 | | |
| DS | DS1 | | 0.769 | 0.778 | 0.798 | | |
| | DS2 | | 0.785 | 0.793 | 0.799 | | |
| | DS3 | | 0.748 | 0.738 | 0.776 | | |
| CS | CS1 | | 0.716 | 0.764 | 0.822 | | |
| | CS2 | | 0.741 | 0.783 | 0.825 | | |
| | CS3 | | 0.710 | 0.738 | 0.806 | | |
| | CS4 | | 0.714 | 0.783 | 0.833 | | |
| | CS5 | | 0.711 | 0.805 | 0.839 | | |
| | CS6 | | 0.729 | 0.785 | 0.814 | | |
| | CS7 | | 0.673 | 0.761 | 0.826 | | |
| | CS8 | | 0.677 | 0.762 | 0.830 | | |
| | CS9 | | 0.703 | 0.774 | 0.827 | | |
| | CS10 | | 0.685 | 0.799 | 0.841 | | |
| | CS11 | | 0.680 | 0.767 | 0.828 | | |
| | CS12 | | 0.684 | 0.801 | 0.849 | | |
| OS | OS1 | | 0.663 | 0.814 | 0.868 | | |
| | OS2 | | 0.663 | 0.810 | 0.858 | | |
| | OS3 | | 0.617 | 0.818 | 0.873 | | |
| SEAP | | 78.395 | | | | 0.972 | |
| | SEAP1 | | 0.639 | 0.742 | 0.781 | | |
| | SEAP2 | | 0.677 | 0.774 | 0.787 | | |
| | SEAP3 | | 0.657 | 0.748 | 0.776 | | |
| | SEAP4 | | 0.675 | 0.778 | 0.787 | | |
| | SEAP5 | | 0.635 | 0.750 | 0.795 | | |
| | SEAP6 | | 0.675 | 0.824 | 0.828 | | |
| | SEAP7 | | 0.678 | 0.812 | 0.817 | | |
| | SEAP8 | | 0.623 | 0.748 | 0.802 | | |
| | SEAP9 | | 0.682 | 0.800 | 0.807 | | |
| | SEAP10 | | 0.658 | 0.756 | 0.786 | | |
| | SEAP11 | | 0.660 | 0.776 | 0.799 | | |
| SECM | | 82.806 | | | | 0.970 | |
| | SECM1 | | 0.682 | 0.776 | 0.735 | | |
| | SECM2 | | 0.686 | 0.775 | 0.728 | | |
| | SECM3 | | 0.728 | 0.849 | 0.757 | | |
| | SECM4 | | 0.715 | 0.843 | 0.745 | | |
| | SECM5 | | 0.720 | 0.827 | 0.747 | | |
| | SECM6 | | 0.724 | 0.856 | 0.763 | | |
| | SECM7 | | 0.685 | 0.812 | 0.768 | | |
| | SECM8 | | 0.712 | 0.838 | 0.750 | | |

**Table 4. Construct Measurement and Convergent Validity.**

| Variables | | Construct/ Items | Loading | AVE | CR |
|---|---|---|---|---|---|
| MSR | IS | IS1: During the project launch period, the government provided a forecast report on the economic benefits that the project would bring to the local area | 0.700 | 0.710 | 0.984 |
| | | IS2: During the project launch period, the government analyzed the feasibility of the technical difficulties of the project | 0.785 | | |
| | | IS3: During the project initiation phase, the government considers the environmental and ecological impact of the project, and has the environmental assessment report (EIA) | 0.780 | | |
| | | IS4: In the project establishment stage, the government considers the situation that the project respects religion, nationality and culture (such as alms giving) | 0.748 | | |
| | | IS5: During the project establishment phase, the media fairly reported the legitimacy of the publicity activities related to the project | 0.803 | | |
| | | IS6: During the project establishment stage, the media always paid attention to the ethical and environmental issues related to the project | 0.773 | | |
| | | IS7: During the project initiation stage, the media paid attention to the needs of the community and the public | 0.794 | | |
| | DS | DS1: In the design stage of the project, the designer adopts the economical optimal design concept under the condition of ensuring the design quality | 0.828 | | |
| | | DS2: In the design stage of the project, the designer considers the project's innovation and technological progress | 0.826 | | |
| | | DS3: During the design phase of the project, the Government actively listened to the public's suggestions on the design scheme | 0.810 | | |
| | CS | CS1: During the construction phase of the project, the project legal person ensures the security of funds and reasonable returns | 0.861 | | |
| | | CS2: In the construction phase of the project, the project legal person adopts the concept of green construction | 0.866 | | |
| | | CS3: During the construction phase of the project, the project legal person shall pay active attention to the needs of the surrounding communities and the public | 0.842 | | |
| | | CS4: During the construction phase of the project, the contractor innovates and improves the construction technology | 0.872 | | |
| | | CS5: During the construction phase of the Project, the Contractor complied with laws, regulations and industry norms | 0.888 | | |
| | | CS6: During the construction phase of the Project, the Contractor ensured the quality of the Works and construction safety | 0.872 | | |
| | | CS7: During the construction phase of the project, the contractor took measures to protect the ecological environment of the surrounding communities and areas | 0.866 | | |
| | | CS8: During the construction phase of the project, the contractor actively deals with emergency public events during construction | 0.876 | | |
| | | CS9: During the construction phase of the project, the supervisor shall supervise the rights and interests of the project construction staff | 0.878 | | |
| | | CS10: During the construction phase of the project, the supervisor shall supervise the environmental protection measures of the project | 0.895 | | |
| | | CS11: In the construction phase of the project, the supplier shall guarantee the quality of construction materials | 0.881 | | |
| | | CS12: In the construction phase of the project, the supplier actively promotes and uses green materials | 0.900 | | |
| | OS | OS1: During the operation phase of the project, the operator carries out routine maintenance on the project | 0.899 | | |
| | | OS2: During the operation phase of the project, the operator shall comply with laws, industry norms and contract provisions | 0.897 | | |
| | | OS3: In the project operation stage, the operator has to protect the community and regional ecological environment | 0.886 | | |

*(Continued)*

**Table 4.** (Continued)

| Variables | Construct/ Items | Loading | AVE | CR |
|---|---|---|---|---|
| SEAP | SEAP1: There are concerns about the use of environmentally friendly building materials by contractors/subcontractors | 0.834 | 0.742 | 0.969 |
| | SEAP2: There is concern that the contractor/subcontractor will agree to place the garbage dump in the designated location | 0.840 | | |
| | SEAP3: There is concern about whether the contractor/subcontractor has cleaned the transport vehicles before driving on the road | 0.828 | | |
| | SEAP4: There are concerns about whether people involved in the project regularly participate in environmental protection education | 0.853 | | |
| | SEAP5: There are concerns about whether the government has targeted garbage bins in various areas | 0.846 | | |
| | SEAP6: There are concerns about whether the government has formulated and implemented environmental protection policies and measures. In this process, I can learn a lot of environmental protection knowledge | 0.910 | | |
| | SEAP7: There are concerns about whether the government media actively publicize the importance of environmental protection, and put forward my own suggestions on environmental protection to the project | 0.895 | | |
| | SEAP8: There are concerns about whether the media have publicly praised enterprises or individuals who have done well in environmental protection measures | 0.853 | | |
| | SEAP9: Pay attention to whether the project legal person and contractor actively adopt green design, such as solar energy and wind energy, etc. | 0.888 | | |
| | SEAP10: Give your opinion to the people involved in the project about what they are doing to protect the environment | 0.850 | | |
| | SEAP11: There are concerns about people consciously classify garbage and do not litter | 0.875 | | |
| SECM | SECM1: The public consciously protects environmental hygiene | 0.873 | 0.801 | 0.970 |
| | SECM2: Environmental experts actively promote environmental protection activities | 0.857 | | |
| | SECM3: The enterprise shall improve the production process and manufacturing technology to reduce environmental pollution | 0.923 | | |
| | SECM4: Companies improve energy efficiency | 0.911 | | |
| | SECM5: An investment and financing system with government investment as the main component and the whole society supporting ecological environment construction shall be established | 0.897 | | |
| | SECM6: The government supports the production and use of alternative energy sources | 0.912 | | |
| | SECM7: The government formulates tax policies for environmental protection | 0.890 | | |
| | SECM8: Scientific and theoretical support provided by researchers for improving environmental compensation measures | 0.896 | | |

Note: Model fitting index: $x^2/df = 2.569$, $p = 0.00$, TLI = 0.894, CFI = 0.902, IFI = 0.903, NFI = 0.850, RMSEA = 0.068.

**Table 5. Discriminant validity.**

| | MSR | SEAP | SECM |
|---|---|---|---|
| MSR | 0.710 | | |
| SEAP | 0.676*** | 0.742 | |
| SECM | 0.508*** | 0.507*** | 0.772 |
| AVE sqr | 0.843 | 0.861 | 0.879 |

***represents p value less than 0.001.

The diagonal is the extraction of AVE evaluation variance variation.

**Table 6. Pearson correlation matrix.**

| Variable | Mean | Standard Deviation | MSR | SEAP | SECM | MSR-IS | MSR-DS | MSR-CS | MSR-OS |
|---|---|---|---|---|---|---|---|---|---|
| MSR | 3.736 | 0.907 | 1 | | | | | | |
| SEAP | 3.693 | 0.925 | 0.848** | 1 | | | | | |
| SECM | 3.588 | 0.816 | 0.697** | 0.728** | 1 | | | | |
| MSR-IS | | | | 0.766** | 0.648** | 1 | | | |
| MSR-DS | | | | 0.746** | 0.652** | | 1 | | |
| MSR-CS | | | | 0.833** | 0.671** | . | | 1 | |
| MSR-OS | | | | 0.870** | 0.695** | | | | 1 |

Note: ** represents correlation is significant at the 0.01 level (2-tailed). MSR-IS, MSR-DS, MSR-CS, MSR-OS is the different lifecycle stage of MSR.

outcomes and stakeholder behaviors. Furthermore, sustained MSR appears to be crucial for the sustainable development of megaprojects, particularly in international contexts, as it helps gain local recognition and support, thereby enhancing the project's competitiveness and social license to operate.

**Regression analysis.** Prior to testing the hypotheses, variance inflation factor (VIF) scores were examined to assess potential multicollinearity. All VIF values were well below the common threshold of 10 [22], indicating that multicollinearity was not a significant issue in the regression models.

Hierarchical regression analysis was employed to test the direct effects hypothesized in H1 and its sub-hypotheses (H1a-H1d). The results are presented in Table 7. Model 1 tested the effect of overall MSR on SECM, while Model 2 tested the effects of the four MSR stages simultaneously. Both models controlled for religion, years of work experience, education level, familiarity with social responsibility, and the respondent's role in the project. As shown in Table 7 (Model 1), overall MSR had a significant positive effect on SECM ($\beta = 0.622$, $p < .001$), thus supporting H1. The results for Model 2 indicated that MSR-DS ($\beta = 0.195$, $p < .001$) and MSR-OS ($\beta = 0.470$, $p < .001$) significantly and positively influenced SECM, supporting H1b and H1d. However, the effects of MSR-IS ($\beta = 0.071$, $p > .05$) and MSR-CS ($\beta = -0.114$, $p > .05$) on SECM were not statistically significant. Therefore, H1a and H1c were not supported. This pattern of results suggests that while overall MSR is important, its direct impact on SECM is primarily driven by responsible practices during the design and operational stages of the megaproject life cycle.

Mediation analysis was conducted using the PROCESS Macro for SPSS (Model 4) with 5,000 bootstrap samples to test H4 and its sub-hypotheses (H4a-H4d) [65]. This analysis assessed the mediating role of SEAP in the relationships between MSR (and its individual stages) and SECM. The results of the mediation analysis are presented in Table 8. For all paths (MSR and each stage), the direct effects on SECM remained significant even after including SEAP in the model. Furthermore, the 95% bias-corrected confidence intervals for all indirect effects (via SEAP) did not include zero. This pattern indicates that SEAP partially mediates these relationships. Specifically, for the overall MSR model, both the direct effect ($\beta = 0.246$, 95% CI [0.114, 0.408]) and the indirect effect ($\beta = 0.376$, 95% CI [0.238, 0.496]) were significant, supporting H4. Similarly, SEAP partially mediated the relationship between each MSR stage and SECM. The direct and indirect effects were all significant: for MSR-IS (direct: $\beta = 0.183$, indirect: $\beta = 0.375$), MSR-DS (direct: $\beta = 0.201$, indirect: $\beta = 0.346$), MSR-CS (direct: $\beta = 0.177$, indirect: $\beta = 0.402$), and MSR-OS (direct: $\beta = 0.213$, indirect: $\beta = 0.369$). For all these effects, the 95% bias-corrected confidence intervals for both the direct and indirect paths did not include zero, providing support for H4a, H4b, H4c, and H4d. Table 8 also shows the proportion of the total effect that was mediated by SEAP. This mediating effect accounted for 60.45% of the total effect of overall MSR on SECM. For the individual stages, the proportional mediation effects ranged from 30.54% (MSR-CS) to 69.46% (MSR-IS). This finding underscores that SEAP serves as a crucial mechanism through which both overall MSR and its stage-specific implementations exert their positive influence on SECM. It suggests that while MSR has a direct impact, engaging stakeholders in environmental activities significantly

amplifies this effect, underscoring the importance of active involvement for translating social responsibility efforts into tangible satisfaction.

## Discussion

Grounded in life cycle, stakeholder, and participation theories, this study empirically investigated the effects of international megaproject social responsibility (MSR) on stakeholder satisfaction with the environmental compensation mechanism (SECM), as well as the mediating role of stakeholder environmental activities participation (SEAP). The analysis utilized survey data from international megaprojects involving Chinese contractors in Thailand. The findings elucidate the pathways through which MSR, across different life cycle stages, influences stakeholder satisfaction, and they clarify the mechanism by which stakeholder participation facilitates compensation satisfaction. The theoretical and practical implications of these findings are discussed below.

## Theoretical contributions

First, this study contributes to theoretical integration by combining life cycle, stakeholder, and participation theories to conceptualize and empirically test the relationships among MSR, SEAP, and SECM. Whereas previous research has often examined these constructs in isolation, the empirical analysis presented here reveals their complex interconnections, thereby enriching the theoretical understanding of social responsibility, stakeholder interaction, and their resultant outcomes in the megaproject context. Specifically, this research extends the boundaries of MSR scholarship by moving beyond its general impact on project performance or social acceptance to investigate its influence mechanisms on a specific and critical outcome: environmental compensation satisfaction. By deconstructing MSR across different project life cycle stages, the study offers a more nuanced perspective on how MSR dynamically impacts stakeholder perceptions throughout the project's progression. This approach aligns with Zeng et al. [6], who also emphasized the need to integrate multiple theoretical lenses in MSR analysis. However, our study differs by operationalizing this integration within a transnational empirical setting, thereby testing the theoretical interconnections under more complex cultural and institutional conditions.

Second, this study reveals the differentiated effects of MSR across various life cycle stages on SECM, thereby deepening the understanding of life cycle theory's application in MSR research. The findings indicate that MSR at the design stage (H1b supported) and operation stage (H1d supported) has a direct positive influence on SECM, whereas MSR at the initiation stage (H1a not supported) and construction stage (H1c not supported) did not exhibit a significant direct effect. A possible explanation is that MSR during the design stage (e.g., conducting comprehensive early EIAs, designing scientifically sound and context-sensitive compensation plans) establishes the "inherent quality" of the compensation scheme, directly shaping stakeholders' initial perceptions of its reasonableness. MSR during the operation stage (e.g., implementing continuous environmental monitoring, adhering to operational standards, fulfilling long-term commitments) influences perceptions of compensation effectiveness over the long term and shapes the proponent's image as a sustainably responsible entity, thereby maintaining and enhancing satisfaction. In contrast, the influence of MSR during the initiation and construction stages likely occurs primarily indirectly through mediating mechanisms like SEAP. Their direct effects may be overshadowed by other factors, such as the short-term disturbances inherently associated with construction activities. This finding underscores that responsible actions at different stages have distinct focal points and mechanisms of influence, thereby providing more refined theoretical guidance for future MSR research that adopts a life cycle perspective. Similar stage-specific effects were noted by Liu and Wang [7] in their study of ecological compensation for infrastructure projects in China. However, their study found a more prominent role for the initiation stage, a difference likely attributable to stronger domestic regulatory frameworks and more established practices of early stakeholder mobilization in that context. Our results suggest that in cross-border settings, the later stages (design and operation) may be more influential for SECM, potentially because they provide more sustained and tangible opportunities for stakeholder engagement.

**Table 7. Influence of MSR, 4 stages of MSR and SEAP on SECM.**

| Variables | Model 1 | | Model 2 | | Model 3 | | Model 4 | | Model 5 | | Model 6 | |
|---|---|---|---|---|---|---|---|---|---|---|---|---|
| | SECM | | SECM | | SECM | | SECM | | SECM | | SECM | |
| | β | s.e. | β | s.e. | β | s.e. | β | s.e. | β | s.e. | β | s.e. |
| MSR | 0.622*** | 0.033 | — | — | — | — | — | — | — | — | — | — |
| MSR-IS | — | — | 0.071 | 0.078 | — | — | 0.183*** | 0.046 | — | — | — | — |
| MSR-DS | — | — | 0.195*** | 0.083 | — | — | — | — | 0.201*** | 0.043 | — | — |
| MSR-CS | — | — | −0.114 | 0.110 | — | — | — | — | — | — | 0.177*** | 0.054 |
| MSR-OS | — | — | 0.470*** | 0.092 | — | — | — | — | — | — | — | — |
| SEAP | — | — | — | — | — | — | 0.502 | 0.047 | 0.487 | 0.046 | 0.495 | 0.055 |
| religions | 0.006 | 0.017 | 0.004 | 0.015 | −0.009 | 0.017 | −0.009 | 0.017 | −0.009 | 0.016 | −0.010 | 0.017 |
| working exp | −0.046 | 0.026 | −0.022 | 0.022 | −0.026 | 0.024 | −0.026 | 0.024 | −0.024 | 0.026 | −0.027 | 0.024 |
| highest edu | 0.019 | 0.032 | 0.030 | 0.028 | 0.042 | 0.030 | 0.042 | 0.030 | 0.035 | 0.030 | 0.038 | 0.030 |
| familiar with SR | 0.010 | 0.032 | 0.023 | 0.028 | −0.003 | 0.031 | −0.003 | 0.031 | 0.007 | 0.030 | 0.004 | 0.031 |
| roles in project | 0.010 | 0.009 | 0.007 | 0.008 | 0.007 | 0.009 | 0.007 | 0.009 | 0.008 | 0.008 | 0.011 | 0.009 |
| Adjusted R2 | 0.481 | | 0.500 | | 0.619 | | 0.554 | | 0.560 | | 0.548 | |
| F | 62.421*** | | 45.067*** | | 105.951*** | | 69.167*** | | 70.974*** | | 67.652*** | |

Note: ***represents p value less than 0.001.

**Table 8. Mediating effect of SEAP between MSR and SECM.**

| | Effect | BootSE | BootLLCI | BootULCI | Effecting Rate |
|---|---|---|---|---|---|
| SEAP Mediating effect | 0.376 | 0.065 | 0.238 | 0.496 | 60.45% |
| MSR Direct effect | 0.246 | 0.075 | 0.114 | 0.408 | 39.55% |
| Total effect | 0.623 | 0.045 | 0.534 | 0.712 | |
| SEAP Mediating effect | 0.375 | 0.053 | 0.274 | 0.484 | 67.22% |
| IS Direct effect | 0.183 | 0.058 | 0.069 | 0.296 | 32.78% |
| Total effect | 0.557 | 0.044 | 0.471 | 0.641 | |
| SEAP Mediating effect | 0.346 | 0.049 | 0.253 | 0.442 | 63.28% |
| DS Direct effect | 0.201 | 0.055 | 0.095 | 0.311 | 36.72% |
| Total effect | 0.547 | 0.043 | 0.460 | 0.629 | |
| SEAP Mediating effect | 0.402 | 0.055 | 0.291 | 0.507 | 69.46% |
| CS Direct effect | 0.177 | 0.063 | 0.062 | 0.306 | 30.54% |
| Total effect | 0.578 | 0.043 | 0.493 | 0.661 | |
| SEAP Mediating effect | 0.369 | 0.067 | 0.230 | 0.497 | 63.48% |
| OS Direct effect | 0.213 | 0.075 | 0.072 | 0.366 | 36.52% |
| Total effect | 0.582 | 0.041 | 0.500 | 0.660 | |

Note: BootSE, BootLLCI and BootULCI are the standard error, lower limit and upper limit of the 95% confidence interval estimated by the positive percentile Bootstrap method, respectively. All values are rounded to three decimal places.

Furthermore, this study provides robust empirical support for the mediating role of stakeholder environmental activities participation (SEAP) in the relationship between MSR and SECM (H4, H4a-H4d supported). This finding validates and enriches the application of participation theory in the field of environmental governance. The results indicate that a high level of MSR promotes stakeholder participation in environmental activities (H2, H2a-H2d supported), which in turn enhances their satisfaction with the environmental compensation mechanism (H3 supported). SEAP, as a mediator, helps to explain the critical

| Model 7 | | Model 8 | | Model 9 | | Model 10 | | Model 11 | | Model 12 | |
| SECM | | SEAP | | SEAP | | SEAP | | SEAP | | SEAP | |
| β | s.e. | β | s.e. | β | s.e. | β | s.e. | β | s.e. | β | s.e. |
|---|---|---|---|---|---|---|---|---|---|---|---|
| — | — | — | — | — | — | — | — | — | — | — | — |
| — | — | 0.746*** | 0.032 | — | — | — | — | — | — | — | — |
| — | — | — | — | 0.710*** | 0.032 | — | — | — | — | — | — |
| — | — | — | — | — | — | 0.812*** | 0.027 | — | — | — | — |
| 0.213*** | 0.058 | — | — | — | — | — | — | 0.821*** | 0.024 | — | — |
| 0.450 | 0.062 | — | — | — | — | — | — | — | — | — | — |
| −0.007 | 0.017 | 0.026 | 0.018 | 0.023 | 0.018 | 0.020 | 0.015 | 0.030 | 0.014 | 0.003 | 0.015 |
| −0.330 | 0.024 | −0.025 | 0.026 | −0.018 | 0.027 | −0.023 | 0.022 | −0.043 | 0.020 | −0.024 | 0.022 |
| 0.040 | 0.030 | −0.044 | 0.032 | −0.069 | 0.033 | −0.061 | 0.028 | −0.040 | 0.025 | 0.026 | 0.027 |
| −0.001 | 0.031 | 0.039 | 0.033 | 0.081 | 0.033 | 0.051 | 0.028 | 0.021 | 0.025 | 0.010 | 0.028 |
| 0.013 | 0.009 | −0.016 | 0.009 | −0.009 | 0.009 | 0.002 | 0.008 | 0.008 | 0.007 | 0.006 | 0.008 |
| 0.551 | | 0.602 | | 0.576 | | 0.704 | | 0.765 | | 0.638 | |
| 68.435*** | | 98.496*** | | 88.620*** | | 154.853*** | | 211.687*** | | 85.808 | |

mechanism—the "black box"—of how MSR influences SECM. Through participation, stakeholders can gain a better understanding of project environmental impacts and compensation measures, voice their concerns and expectations, and even co-participate in decision-making and monitoring processes. This participatory process enhances the transparency, perceived fairness, and responsiveness of the compensation process. It builds trust between project proponents and stakeholders, making them more likely to accept and be satisfied with the final outcomes. Consistent with the perspectives of Kaku and Zhu [70] and Ma and Sun [19] on the significance of stakeholder engagement, this study further demonstrates that, within the MSR framework, facilitating stakeholder environmental participation constitutes an effective pathway to enhance environmental compensation satisfaction. However, unlike Kaku et al. [70], who reported a full mediation effect in environmental governance projects, our findings indicate partial mediation. This suggests that other contextual factors—such as cultural norms, governance structures, and prior project experiences—may also play significant roles in shaping stakeholder satisfaction.

Finally, by examining international megaprojects involving Chinese contractors in Thailand, this research contributes valuable empirical evidence to the literature on cross-cultural megaproject management and social responsibility. Testing these relationships within a specific socio-cultural and institutional context helps to reveal both the applicability of the theoretical model and its potential boundary conditions in diverse environments. This finding holds significant theoretical implications for advancing the generalizability of MSR theory and for informing its application in transnational projects, such as those under the Belt and Road Initiative [2]. Whereas prior studies (e.g., Ma et al. [16]) have examined MSR primarily in domestic contexts, our research expands the empirical base to a cross-cultural setting. It highlights both consistencies—such as the positive role of stakeholder engagement—and divergences, particularly regarding the life cycle stages that exert the strongest effects. This comparative perspective underscores the importance of adapting MSR strategies to specific cultural and institutional environments.

## Practical implications

The findings of this study offer important practical implications for international megaproject managers and relevant policymakers, particularly particularly those aimed at enhancing the satisfaction with environmental compensation mechanisms.

The research emphasizes that proponents of international megaprojects should integrate social responsibility practices throughout the entire project life cycle, but with a differentiated strategic focus depending on the stage. Given that MSR at the design stage exerts a direct influence on SECM, project proponents should invest substantial resources and effort early in the process. This includes conducting high-quality environmental impact assessments (EIAs) and proactively inviting stakeholders to participate fully in the co-design of compensation schemes. Such participation helps ensure the scientific validity, practical reasonableness, and social acceptability of the plans, serving as a crucial upfront strategy for enhancing eventual satisfaction. Similarly, as MSR during the operation stage also directly influences satisfaction, project proponents cannot disengage after project delivery. Instead, they must establish long-term mechanisms for effective environmental monitoring and transparent information disclosure. They must also continuously fulfill all environmental compensation commitments and maintain open, regular communication channels with local communities to maintain and enhance long-term satisfaction.

This study strongly underscores the strategic importance of actively facilitating stakeholder environmental activities participation (SEAP). Project proponents should deliberately create opportunities and platforms to encourage and support key stakeholders (e.g., local communities, NGOs) to participate meaningfully in environmental impact assessments, in the development and monitoring of environmental management plans, and in the implementation and evaluation of environmental compensation schemes. Specifically, project proponents should prioritize significant increases in information transparency. This involves proactively and timely disclosing project environmental information, EIA reports, compensation plans, and implementation progress in formats that are easily understandable to diverse stakeholders. Concurrently, multi-channel communication mechanisms should be established. These could include appointing dedicated community liaison officers, holding regular community communication meetings, and creating accessible online feedback platforms. The goal is to ensure that stakeholders' voices are not only heard but also acknowledged and addressed effectively. Furthermore, efforts should shift from mere consultation toward substantive participation, such as co-design and co-monitoring initiatives. Practical steps could include inviting community representatives to join independent environmental monitoring teams or involving them in joint decision-making processes for allocating compensation funds. Providing necessary training and accessible information support to stakeholders is also crucial to empower their effective participation. These actions can help bridge information gaps, enhance stakeholders' trust in the project's environmental management, improve the perceived fairness and acceptability of compensation schemes, and ultimately lead to a significant increase in environmental compensation satisfaction.

Moreover, the results of this study offer insights for other stakeholders, including government agencies, NGOs, and investors. Government agencies should formulate and strengthen regulations to mandate meaningful stakeholder participation in the environmental governance of large projects, including compensation processes. They should also provide clear guidance and robust oversight to ensure that social responsibility commitments are effectively implemented. NGOs can leverage their role as communication bridges between projects and communities. They can assist in organizing effective community participation and conduct independent monitoring of project environmental performance, thereby fostering more transparent dialogue and constructive cooperation. For investors and financial institutions, the project proponent's track record in MSR performance throughout the life cycle and its stakeholder engagement efforts should be incorporated as critical indicators in risk and sustainability assessments when evaluating large project investments. Prioritizing investments in projects that demonstrate robust MSR and stakeholder engagement can not only mitigate environmental and social risks but also contribute to obtaining a stronger social license to operate and enhance long-term project viability and success. In conclusion, by strategically fulfilling MSR and actively facilitating SEAP, international megaprojects can more effectively enhance satisfaction with environmental compensation mechanisms. This approach supports the achievement of economic objectives while simultaneously promoting sustainable development and building a positive reputation in host countries. The practical strategies outlined in this study provide valuable guidance for managing MSR in international

engineering projects, offering particular relevance for Chinese enterprises engaged in "Going Global" and "Belt and Road" initiatives [16,23].

## Limitations and future directions

Despite its contributions, this study has several limitations that should be addressed in future research, and which also open avenues for further exploration. First, the reliance on self-reported survey data introduces the potential for common method bias and other subjective biases, as responses are contingent on individual perceptions and recall. Second, the geographical focus on a specific context (Thailand) may limit the generalizability of the findings. Cultural, economic, and legal frameworks can significantly shape the dynamics of MSR and SECM, suggesting that the results might not be directly transferable to other regions or countries with different institutional environments. Third, SECM is likely influenced by a multitude of factors beyond the scope of this study, such as macroeconomic trends, broader political landscapes, technological advancements, and exogenous shocks (e.g., natural disasters, pandemics). These factors were not controlled for in the current analysis. Future research could address these limitations by incorporating more objective data sources (e.g., archival data on compensation outcomes), expanding the geographical and cultural scope of study, and including a broader array of control and contextual variables that might influence SECM.

## Conclusion

Grounded in project life cycle, stakeholder, and participation theories, this study empirically investigated the relationships between international megaproject social responsibility (MSR), stakeholder environmental activities participation (SEAP), and satisfaction with the environmental compensation mechanism (SECM). The findings significantly expand the discourse on MSR within international contexts and enhance our understanding of the key factors driving SECM.

The results reveal that MSR exerts a positive influence on environmental compensation satisfaction, with differentiated effects across various project life cycle stages. Specifically, MSR during the design and operation stages demonstrates a direct positive impact on satisfaction. Crucially, the study confirms that MSR promotes stakeholder engagement in environmental activities (SEAP), and that this participation, in turn, significantly enhances satisfaction with environmental compensation mechanisms (SECM). SEAP was identified as a pivotal mediator in the relationship between MSR and SECM. This suggests that by meaningfully involving stakeholders in environmental initiatives, megaprojects can effectively improve local residents' overall satisfaction with environmental compensation.

This research offers novel theoretical insights into MSR, primarily by illuminating the core mechanism—stakeholder participation—through which MSR influences environmental compensation satisfaction. On a practical level, the findings provide valuable strategic recommendations for international contractors, local governments, and other stakeholders. They underscore the critical importance of proactively fulfilling social responsibility and fostering stakeholder participation throughout the megaproject life cycle—particularly during the design and operation stages—to develop effective sustainable development strategies and enhance public satisfaction in host countries.

## Supporting information

**S1 File. A spreadsheets files containing scripts datasets in this study.**
(XLSX)

**S2 File. Questionnaire used for data collection in this study.**
(DOCX)

**S3 File. Inclusivity-in-global-research-questionnaire.**
(DOCX)

## Author contributions

**Conceptualization:** Zixuan Zeng.

**Data curation:** Zixuan Zeng, Yongtao Shen.

**Formal analysis:** Zixuan Zeng, Yongtao Shen.

**Funding acquisition:** Zixuan Zeng.

**Investigation:** Zixuan Zeng.

**Methodology:** Jijun Yang.

**Project administration:** Thammanoon Hengsadeekul.

**Resources:** Thammanoon Hengsadeekul.

**Writing – original draft:** Zixuan Zeng.

**Writing – review & editing:** Zixuan Zeng.

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
