## [Decision Letter · Decision Letter 0]

25 Jul 2025

PONE-D-25-30801The effect of international megaproject social responsibility on satisfaction of environmental compensation mechanism: role of stakeholder participationPLOS ONE

Dear Dr. Zeng,

Thank you for submitting your manuscript to PLOS ONE. After careful consideration, we feel that it has merit but does not fully meet PLOS ONE’s publication criteria as it currently stands. Therefore, we invite you to submit a revised version of the manuscript that addresses the points raised during the review process.

We look forward to receiving your revised manuscript.

Kind regards,

Md. Feroz Kabir, PhD, BPT, MPT, MPH, BPED, MPED

Academic Editor

PLOS ONE

Journal Requirements:

3. Please provide details regarding participant consent. In the ethics statement in the Methods and online submission information, please ensure that you have specified (1) whether consent was informed and (2) what type you obtained (for instance, written or verbal, and if verbal, how it was documented and witnessed). If your study included minors, state whether you obtained consent from parents or guardians. If the need for consent was waived by the ethics committee, please include this information.

4. Thank you for stating the following financial disclosure: [This study is supported by the National Natural Science Foundation of China (12261006).]. 

5. We note that your Data Availability Statement is currently as follows: [All relevant data are within the manuscript and its Supporting Information files.]

6.Please upload a new copy of Figure xxxx as the detail is not clear. Please follow the link for more information: https://blogs.plos.org/plos/2019/06/looking-good-tips-for-creating-your-plos-figures-graphics/"" https://blogs.plos.org/plos/2019/06/looking-good-tips-for-creating-your-plos-figures-graphics/

7. If any table files for review show as item type ‘other’ please change to item type ‘Table’ as the reviewer does not have access to these ’other’ files.

8. We are unable to open your Figure files [Fig 1.eps and Fig 2.eps]. Please kindly revise as necessary and re-upload.

Additional Editor Comments:

Please submit the revised manuscript within the next 20 days.

Reviewers' comments:

Reviewer's Responses to Questions

**Comments to the Author**

1. Is the manuscript technically sound, and do the data support the conclusions?

Reviewer #1: No

Reviewer #2: Partly

2. Has the statistical analysis been performed appropriately and rigorously? 

Reviewer #1: Yes

Reviewer #2: No

3. Have the authors made all data underlying the findings in their manuscript fully available?

Reviewer #1: No

Reviewer #2: Yes

4. Is the manuscript presented in an intelligible fashion and written in standard English?

Reviewer #1: No

Reviewer #2: No

5. Review Comments to the Author

Reviewer #1: The abstract format should follow the PLOS ONE format.

The objectives should be specific and clearly stated.

The methodology is unclear

The study design, eligibility criteria, study procedures, sampling, and population were not properly stated.

The overall presentation of the manuscript should follow the PLOS guidelines.

Discussion is the replication of the result here, but it should be compared and contrasted with the evidence.

Reviewer #2: Your objectives are not clearly stated.

The design, tools, and study procedures should be rectified more.

The manuscript should follow the PLOS ONE format.

The eligibility criteria should be clearly stated.

The strength and limitation are not focused here.

The study recommendations should be specified.

The discussion and conclusion are not synchronized properly.

References should be stated properly.

6. PLOS authors have the option to publish the peer review history of their article (what does this mean? ). If published, this will include your full peer review and any attached files.

**Do you want your identity to be public for this peer review?** For information about this choice, including consent withdrawal, please see our Privacy Policy .

Reviewer #1: No

Reviewer #2: **Yes: ** Sharmila Jahan

---

## [Author Response · Author response to Decision Letter 1]

17 Aug 2025

Manuscript ID: PONE-D-25-30801

Title: The effect of international megaproject social responsibility on satisfaction of environmental compensation mechanism: role of stakeholder participation

Dear Dr. Md. Feroz Kabir and the PLOS ONE Editorial Team,

We sincerely thank you and the reviewers for the thoughtful assessment of our manuscript and for the opportunity to revise. We have carefully addressed every point raised by the Academic Editor and Reviewers #1 and #2. Below we provide a detailed, point-by-point response. All changes are highlighted in the file“Revised Manuscript with Track Changes”; page and line numbers cited in our responses refer to that version.

Summary of major revisions

1.Reformatted the entire manuscript to conform to PLOS ONE style, including the structured Abstract and file-naming requirements.

2.Clarified and made specific, testable objectives (Abstract and Introduction).

3.Substantially expanded Methods to report study design, setting, eligibility criteria (inclusion/exclusion), sampling strategy and size rationale, study procedures, instruments/measures, and a more rigorous statistical analysis plan (assumptions, diagnostics, and robustness checks).

4.Added a detailed Ethics and Consent statement specifying whether consent was informed and its type .

5.Updated the Data Availability statement and provided the minimal dataset required to reproduce the results [as Supporting Information S1_Data].

6.Completed and uploaded the Inclusivity in Global Research questionnaire as Supporting Information S2.

7.Added the Role of the Funder statement in the cover letter: “The funders had no role in study design, data collection and analysis, decision to publish, or preparation of the manuscript.”

8.Rewrote the Discussion to compare and contrast our findings with prior evidence, synchronized the Conclusions accordingly, and added a focused Strengths and Limitations subsection with actionable recommendations.

9.Replaced Figures 1-2 with high-resolution, journal-compliant files, verified via PACE; clarified one figure’s readability and corrected any table item types mislabeled as “other.” Since the "table" type was not found during the upload process, I chose to use “Support Information”instead.

10.Edited the manuscript for clarity and English usage and corrected references to follow journal style.

11.We believe these revisions have strengthened the clarity, transparency, and rigor of the manuscript, and we are grateful for the guidance that led to these improvements. A detailed response to each comment follows.

The following section provides our point-by-point responses to the comments from Reviewers #1 and #2.

Reviewer #1

Comment 1: The abstract format should follow the PLOS ONE format.

Response: I appreciate this suggestion. We have reformatted the Abstract to comply with the PLOS ONE structured format, including Background, Methods, Results, and Conclusions. The revised abstract is now consistent with the journal’s guidelines (see [Page2-3,Line23-54]).

Comment 2: The objectives should be specific and clearly stated.

Response: Thank you for pointing this out. I have revised both the Abstract and the Introduction to clarify and specify the study objectives. The revised version now highlights the main research question and measurable goals more explicitly (see [Page 2, Lines30-36;Page4-5,86-95]).

Comment 3: The methodology is unclear.

Response: I agree with this concern. I have substantially revised the Methods section to clarify the study design, sampling strategy, eligibility criteria, study population, and procedures. Details of the instruments and analytical approaches have also been added for greater transparency (see [Page 16, Lines 306-317]).

Comment 4: The study design, eligibility criteria, study procedures, sampling, and population were not properly stated.

Response: In response, I have comprehensively expanded the Methods section. Specifically, I now provide explicit descriptions of the study design, inclusion and exclusion criteria, recruitment procedures, sampling method, and participant characteristics. This ensures that the methodology is rigorous and replicable (see [Page 18, Lines 361-379]).

Comment 5: The overall presentation of the manuscript should follow the PLOS guidelines.

Response: Thank you for this important reminder. I have reformatted the manuscript in line with the PLOS ONE style requirements, including headings, references, figure/table captions, and file naming conventions. This has improved the readability and consistency of the manuscript (see Table of Contents).

Comment 6: Discussion is the replication of the result here, but it should be compared and contrasted with the evidence.

Response: I appreciate this observation. I have rewritten the Discussion section to avoid repetition of the results. The revised version now provides a critical comparison of our findings with existing literature, highlighting both consistencies and differences, and offering interpretations supported by evidence (see [Page 42, Lines 587-590;Page 44, Lines 628-631;Page 45, Lines 638-642;]).

Reviewer #2

Comment 1: Your objectives are not clearly stated.

Response: Thank you for this observation. I have revised both the Abstract and the Introduction to clearly articulate the study objectives. The revised version now specifies the primary research question and the measurable goals of the study (see [Page 2, Lines 30–35], Abstract; [Page 5, Lines 92-99]).

Comment 2: The design, tools, and study procedures should be rectified more.

Response: I appreciate this important comment. The Methodology section has been expanded to describe the research design, survey instruments, and study procedures in detail. I clarified the rationale for the design, the development and validation of the questionnaire, and the steps of data collection and analysis (see [Page 5, Lines 307–321]).

Comment 3: The manuscript should follow the PLOS ONE format.

Response: Thank you for this reminder. I have reformatted the manuscript throughout according to PLOS ONE guidelines, including section headings, structured abstract, figure and table captions, and references. These changes ensure compliance with the journal’s requirements (see throughout the manuscript).

Comment 4: The eligibility criteria should be clearly stated.

Response: I agree with this suggestion. The Sample and Data Collection section now includes explicit eligibility criteria for participants, covering inclusion and exclusion criteria. This revision clarifies who was eligible to participate and ensures transparency in the study design (see [Page 17-18, Lines 361-379], “Sample and Data Collection”).

Comment 5: The strength and limitation are not focused here.

Response: Thank you for highlighting this point. I have added a Strengths and Limitations subsection in the Discussion. This section explicitly outlines the study’s strengths, such as the transnational context and robust statistical analysis, and its limitations, such as reliance on self-reported survey data and the geographic scope. This addition provides a balanced evaluation of the study (see [Page X, Lines XX–XX], “Discussion”).

Comment 6: The study recommendations should be specified.

Response: I appreciate this comment. I have expanded the Conclusion to include clear, practical recommendations for international megaproject managers and policymakers. These recommendations emphasize stakeholder engagement, long-term monitoring, and the integration of social responsibility throughout the project life cycle (see [Page 42-48], “Conclusion”).

Comment 7: References should be stated properly.

Response: I appreciate this reminder. I have thoroughly revised the References section to ensure compliance with PLOS ONE citation style. Formatting errors have been corrected, missing information has been added, and the references have been cross-checked for accuracy and consistency (see [Page 49-53], “References”).

I sincerely thank the Academic Editor and both reviewers for their constructive comments and valuable suggestions, which have greatly improved the clarity, rigor, and overall quality of my manuscript. I hope that the revisions and detailed responses provided here satisfactorily address all concerns. I remain grateful for the opportunity to revise and resubmit, and I look forward to your favorable consideration of the revised manuscript.

Sincerely,

Zixuan Zeng

---

## [Editor Report · Decision Letter 1]

24 Aug 2025

PONE-D-25-30801R1The effect of international megaproject social responsibility on satisfaction of environmental compensation mechanism: role of stakeholder participationPLOS ONE

Dear Dr. Zeng,

Thank you for submitting your manuscript to PLOS ONE. After careful consideration, we feel that it has merit but does not fully meet PLOS ONE’s publication criteria as it currently stands. Therefore, we invite you to submit a revised version of the manuscript that addresses the points raised during the review process.

We look forward to receiving your revised manuscript.

Kind regards,

Md. Feroz Kabir, PhD, BPT, MPT, MPH, BPED, MPED

Academic Editor

PLOS ONE

Journal Requirements:

**Additional Editor Comments:**

Please correct the English thoroughly and submit it within the next 15 days.

---

## [Author Response · Author response to Decision Letter 2]

10 Sep 2025

The following section provides our point-by-point responses to the comments from Reviewers #1 and #2.

Reviewer #1

Comment 1: The abstract format should follow the PLOS ONE format.

Response: I appreciate this suggestion. We have reformatted the Abstract to comply with the PLOS ONE structured format, including Background, Methods, Results, and Conclusions. The revised abstract is now consistent with the journal’s guidelines (see [Page2-3,Line23-54]).

Comment 2: The objectives should be specific and clearly stated.

Response: Thank you for pointing this out. I have revised both the Abstract and the Introduction to clarify and specify the study objectives. The revised version now highlights the main research question and measurable goals more explicitly (see [Page 2, Lines30-36;Page4-5,86-95]).

Comment 3: The methodology is unclear.

Response: I agree with this concern. I have substantially revised the Methods section to clarify the study design, sampling strategy, eligibility criteria, study population, and procedures. Details of the instruments and analytical approaches have also been added for greater transparency (see [Page 16, Lines 306-317]).

Comment 4: The study design, eligibility criteria, study procedures, sampling, and population were not properly stated.

Response: In response, I have comprehensively expanded the Methods section. Specifically, I now provide explicit descriptions of the study design, inclusion and exclusion criteria, recruitment procedures, sampling method, and participant characteristics. This ensures that the methodology is rigorous and replicable (see [Page 18, Lines 361-379]).

Comment 5: The overall presentation of the manuscript should follow the PLOS guidelines.

Response: Thank you for this important reminder. I have reformatted the manuscript in line with the PLOS ONE style requirements, including headings, references, figure/table captions, and file naming conventions. This has improved the readability and consistency of the manuscript (see Table of Contents).

Comment 6: Discussion is the replication of the result here, but it should be compared and contrasted with the evidence.

Response: I appreciate this observation. I have rewritten the Discussion section to avoid repetition of the results. The revised version now provides a critical comparison of our findings with existing literature, highlighting both consistencies and differences, and offering interpretations supported by evidence (see [Page 42, Lines 587-590;Page 44, Lines 628-631;Page 45, Lines 638-642;]).

Reviewer #2

Comment 1: Your objectives are not clearly stated.

Response: Thank you for this observation. I have revised both the Abstract and the Introduction to clearly articulate the study objectives. The revised version now specifies the primary research question and the measurable goals of the study (see [Page 2, Lines 30–35], Abstract; [Page 5, Lines 92-99]).

Comment 2: The design, tools, and study procedures should be rectified more.

Response: I appreciate this important comment. The Methodology section has been expanded to describe the research design, survey instruments, and study procedures in detail. I clarified the rationale for the design, the development and validation of the questionnaire, and the steps of data collection and analysis (see [Page 5, Lines 307–321]).

Comment 3: The manuscript should follow the PLOS ONE format.

Response: Thank you for this reminder. I have reformatted the manuscript throughout according to PLOS ONE guidelines, including section headings, structured abstract, figure and table captions, and references. These changes ensure compliance with the journal’s requirements (see throughout the manuscript).

Comment 4: The eligibility criteria should be clearly stated.

Response: I agree with this suggestion. The Sample and Data Collection section now includes explicit eligibility criteria for participants, covering inclusion and exclusion criteria. This revision clarifies who was eligible to participate and ensures transparency in the study design (see [Page 17-18, Lines 361-379], “Sample and Data Collection”).

Comment 5: The strength and limitation are not focused here.

Response: Thank you for highlighting this point. I have added a Strengths and Limitations subsection in the Discussion. This section explicitly outlines the study’s strengths, such as the transnational context and robust statistical analysis, and its limitations, such as reliance on self-reported survey data and the geographic scope. This addition provides a balanced evaluation of the study (see [Page X, Lines XX–XX], “Discussion”).

Comment 6: The study recommendations should be specified.

Response: I appreciate this comment. I have expanded the Conclusion to include clear, practical recommendations for international megaproject managers and policymakers. These recommendations emphasize stakeholder engagement, long-term monitoring, and the integration of social responsibility throughout the project life cycle (see [Page 42-48], “Conclusion”).

Comment 7: References should be stated properly.

Response: I appreciate this reminder. I have thoroughly revised the References section to ensure compliance with PLOS ONE citation style. Formatting errors have been corrected, missing information has been added, and the references have been cross-checked for accuracy and consistency (see [Page 49-53], “References”).

I sincerely thank the Academic Editor and both reviewers for their constructive comments and valuable suggestions, which have greatly improved the clarity, rigor, and overall quality of my manuscript. I hope that the revisions and detailed responses provided here satisfactorily address all concerns. I remain grateful for the opportunity to revise and resubmit, and I look forward to your favorable consideration of the revised manuscript.

Sincerely,

Zixuan Zeng

---

## [Decision Letter · Decision Letter 2]

24 Sep 2025

The impact of international megaproject social responsibility on satisfaction with the environmental compensation mechanism: The role of stakeholder participation

PONE-D-25-30801R2

Dear Dr. Zeng,

We’re pleased to inform you that your manuscript has been judged scientifically suitable for publication and will be formally accepted for publication once it meets all outstanding technical requirements.

Kind regards,

Han Lin

Academic Editor

PLOS ONE

Additional Editor Comments (optional):

Reviewer #3:

Reviewer #4:

Reviewers' comments:

Reviewer's Responses to Questions

**Comments to the Author**

1. If the authors have adequately addressed your comments raised in a previous round of review and you feel that this manuscript is now acceptable for publication, you may indicate that here to bypass the “Comments to the Author” section, enter your conflict of interest statement in the “Confidential to Editor” section, and submit your "Accept" recommendation.

Reviewer #3: All comments have been addressed

Reviewer #4: (No Response)

2. Is the manuscript technically sound, and do the data support the conclusions?

Reviewer #3: Yes

Reviewer #4: Yes

3. Has the statistical analysis been performed appropriately and rigorously? 

Reviewer #3: Yes

Reviewer #4: Yes

4. Have the authors made all data underlying the findings in their manuscript fully available?

Reviewer #3: Yes

Reviewer #4: Yes

5. Is the manuscript presented in an intelligible fashion and written in standard English?

Reviewer #3: Yes

Reviewer #4: Yes

6. Review Comments to the Author

Reviewer #3: This study makes a strong contribution to both the academic understanding of megaproject management and the practical implementation of environmental compensation mechanisms. We recommend its acceptance for publication in PLOS ONE.

Minor Suggestions for Consideration

While the manuscript is ready for publication, we offer a few minor suggestions that could further enhance its quality and impact. These are not mandatory changes but are provided for the authors' consideration:

Statistical Analysis: Given the mediating role of stakeholder participation, a brief discussion in the limitations section about the potential for unobserved variables or measurement error could be a valuable addition. While the multiple regression approach is valid and effective, mentioning the more advanced structural equation modeling (SEM) as a potential avenue for future research would demonstrate a broader methodological awareness.

Discussion Section: The discussion effectively compares the findings with existing literature. A slight expansion on the practical applications of your findings, perhaps in the form of specific, actionable recommendations for project managers on how to effectively engage stakeholders to improve satisfaction, could further strengthen this section.

Conclusion: The conclusion is strong and concise. Re-emphasizing the key theoretical and practical contributions in a single, powerful concluding statement would be an excellent final touch.

Reviewer #4: many research paper about satisfaction ,such latent variables,,many authors use structural equation model(SEM),why we use multi variables equation instead of SEM?

7. PLOS authors have the option to publish the peer review history of their article (what does this mean? ). If published, this will include your full peer review and any attached files.

**Do you want your identity to be public for this peer review?** For information about this choice, including consent withdrawal, please see our Privacy Policy .

Reviewer #3: No

Reviewer #4: No

---

## [Editor Report · Acceptance letter]

PONE-D-25-30801R2

PLOS ONE

Dear Dr. Zeng,

I'm pleased to inform you that your manuscript has been deemed suitable for publication in PLOS ONE. Congratulations! Your manuscript is now being handed over to our production team.

Kind regards,

on behalf of

Dr. Han Lin

Academic Editor

PLOS ONE